# Soy Protein Isolate Affects Blood and Brain Biomarker Expression in a Mouse Model of Fragile X

**DOI:** 10.3390/ijms26136137

**Published:** 2025-06-26

**Authors:** Brynne Boeck, Yingqing Mao, Ruo-Pan Huang, Cara J. Westmark

**Affiliations:** 1Department of Neurology, University of Wisconsin, Madison, WI 53706, USA; bboeck@wisc.edu; 2RayBiotech, Inc., Peachtree Corners, GA 30092, USA; mike@raybiotech.com (Y.M.); rhuang@raybiotech.com (R.-P.H.); 3Molecular Environmental Toxicology Center, University of Wisconsin, Madison, WI 53706, USA

**Keywords:** *Fmr^KO^* mouse, fragile X syndrome, cytokine, soy diet

## Abstract

Fragile X syndrome is characterized by the diminished expression of the fragile X messenger ribonucleoprotein (FMRP), a ubiquitously expressed RNA binding protein with numerous functions in cells. Our prior work found significant differences in physiological and behavioral outcomes as a function of FMRP levels and in response to diet in mice. Here, we assess protein biomarker levels as a function of FMRP levels, sex and matched casein and soy protein isolate-based purified ingredient diets in *Fmr1^KO^* and littermate mice. Brain regions (cortex, hippocampus, and hypothalamus) and blood plasma were analyzed by RayBiotech’s Quantibody^®^ Mouse Cytokine Antibody Array 640 to quantitate the expression of 640 proteins. The main findings were the identification of numerous proteins that were differentially expressed in response to diet, sex and/or genotype. Of note, prolactin (PRL) levels in blood plasma were significantly elevated in *Fmr1^KO^* female mice as a function of genotype and sex selectively with the AIN-93G/casein diet. Also, using a moderately stringent significance cutoff, growth differentiation factor 9 (GDF-9) in plasma from mice fed AIN-93G/soy was the only protein studied by Quantibody arrays that was differentially expressed between WT and *Fmr1^KO^* male mice. When comparing the results from a pelleted infant formula study with AIN-93G-based diets, insulin-like growth factor binding protein 5 (IGFBP5) in plasma was the only protein differentially expressed as a function of soy in the diet. There was no overlap in statistically significant results when comparing tissue analyzed by mass spectrometry versus Quantibody arrays from mice maintained on AIN-93G-based diets. In conclusion, gene–diet interactions affect protein expression in *Fmr1^KO^* and littermate mice and need to be considered in study design.

## 1. Introduction

There is a poor understanding of how nutrition interacts with genetics to affect development and disease outcomes. We hypothesize that the consumption of high levels of soy protein isolate during postnatal development, through the use of single-source diets such as rodent chows and infant formula, is a dietary exposure that increases the risk of developing seizures, autism and obesity, particularly in vulnerable populations such as those with fragile X syndrome (FXS). FXS is the leading known genetic cause of autism and the most common form of inherited intellectual disability. This X-chromosome disorder is predominantly caused by a CGG trinucleotide repeat expansion in the 5′-untranslated region (UTR) of the *FMR1* gene, which causes the transcriptional silencing of the promoter and the loss of expression of fragile X messenger ribonucleoprotein (FMRP) [1,2]. FMRP binds to RNA and regulates protein synthesis, RNA editing and microRNA, with decreased expression associated with intellectual disability, autistic behaviors and seizures [3,4]. The most widely used preclinical model for the study of FXS is the *Fmr1^KO^* mouse, which lacks the expression of FMRP [5].

Our prior work found altered seizure propensity, body mass, hyperactivity, blood amino acid and phytoestrogen levels, brain proteomics, gut permeability and gut microbial diversity in mice, as well as seizures, autistic behaviors, allergies and gastrointestinal problems in children in response to single-source soy-based diets [6,7,8,9,10,11,12,13,14,15,16,17]. The underlying mechanisms remain to be studied and likely include altered signal transduction and protein expression in response to high circulating levels of soy phytoestrogens [16,18,19].

Screening biomarkers has the potential to identify proteins that are dysregulated in response to diet. Herein, we compared protein expression in three brain regions (cortex, hippocampus, and hypothalamus) as well as blood plasma as a function of *Fmr1* genotype, sex and matched AIN-93G purified ingredient diets containing casein versus soy protein isolate (SPI). This is important because diet is expected to impact developmental and disease outcomes, and diet/drug interactions could have profound effects on preclinical and translational research. Over 120 million mice are used globally per year for preclinical medical research. In 2023, at the University of Wisconsin-Madison, over 100,000 mice were used for research including over 77,000 in the School of Medicine and Public Health (SMPH) where at least 18 different standard diets were employed across vivariums. A quarter of those standard chows contained soy as the protein source. More than 60% of SMPH vivariums used corncob bedding. Soy, corn and wheat are common chow ingredients. It remains to be determined how proteins and other bioactive components in diet and bedding interact with genes and mediate disease pathology.

The aim of this study is to identify proteins with altered expression in response to AIN-93G/soy compared to a matched control diet with casein as the protein source. The main conclusion is that a purified ingredient diet with the sole protein source of SPI alters the expression of several brain and blood proteins as a function of *Fmr1* genotype and sex in mice.

## 2. Results

### 2.1. Differentially Expressed Proteins in Cortex

Several targets were differentially expressed in the cortex as a function of genotype and/or diet in mice. The genotype-specific differences in mice fed AIN-93G/casein included OLFM-1 (array 12) which was decreased 1.6-fold in *Fmr1^HET^* versus *Fmr1^KO^* females (Table 1). CCL6 (array 7) was increased 51-fold in *Fmr1^HET^* females versus *Fmr1^KO^* males. FCRL5 (array 18) was increased in *Fmr1^HET^* females versus both *Fmr1^KO^* females (22-fold) and males (12-fold). IGFBP-2 (3.5-fold), OPG (9.7-fold) and SCF (2.5-fold) (array 4); Eotaxin (3.2-fold), KC (3.3-fold), LEP (31-fold) and MIG (33-fold) (array 5); and CT-1 (2.7-fold) (array 6) were increased in *Fmr1^KO^* females versus *Fmr1^KO^* males mice. BAMBI was increased in WT versus *Fmr1^KO^* males (1.5-fold) (array 19). BMP-7 and FAM3C (array 19) were increased in WT males versus both *Fmr1^KO^* males (6.9-fold and 12-fold, respectively) and females (7.5-fold and 11-fold, respectively). BID and EPHA1 (array 19) were increased at least 34-fold in WT males versus all casein cohorts. CD99-L2 (array 9) was increased 2.3-fold in *Fmr1^KO^* males versus WT males.

Several targets were differentially expressed in the cortex as a function of genotype in mice fed AIN-93G/soy (Table 1). IL-1β (array 5) was increased 2.1-fold in *Fmr1^HET^* females versus WT males. PODXL (array 12) was increased 3.0-fold in *Fmr1^KO^* females compared with *Fmr1^HET^*. CD164 (array 14) was increased 2.4-fold in *Fmr1^KO^* males versus *Fmr1^HET^* females. CST3 (array 7) was decreased 1.3-fold in *Fmr1^KO^* male versus *Fmr1^HET^* female and WT male mice.

In terms of the diet-specific differences comparing protein expression for each genotype maintained on an AIN-93G/casein versus AIN-93G/soy diet, 18 proteins were differentially expressed in the cortex (Table 2). Of note, the differences were specific to a single genotype with B7-2, CD14, OLFM-1, CAMK4 and FCRL5 differences in *Fmr1^HET^* females; OPG, SCF, Eotaxin, LEP, MIG, CT-1 and IL-3 Rb differences in *Fmr1^KO^* females; GHR, BID, BMP-7, EPHA1 and FAM3C differences in WT males; and CD164 differences in *Fmr1^KO^* males.

### 2.2. Differentially Expressed Proteins in Hippocampus

Several targets were differentially expressed in the hippocampus as a function of genotype and/or diet in mice. The genotype-specific differences in mice fed AIN-93G/casein included increased CHST4 11-fold (array 14) and SPB10 10-fold (array 16) in *Fmr1^HET^* females versus *Fmr1^KO^* males (Table 3). MMP-10 (array 8) was decreased 5.4-fold in *Fmr1^HET^* females compared with *Fmr1^KO^* males. TRANCE (array 8) and PRSS34 (array 12) were increased more than 7-fold in *Fmr1^HET^* females compared with both female and male *Fmr1^KO^*. IL-20 Rβ and LIF (array 10) were increased 9- and 3.7-fold, respectively, in *Fmr1^KO^* females compared with *Fmr1^KO^* males. IL-15 Rα and KLB (array 10) were increased over 4-fold in *Fmr1^KO^* females compared with WT and *Fmr1^KO^* males. CD45 1.9-fold (array 9) and EGF 5.6-fold (array 4) were increased in WT males compared with *Fmr1^KO^* females. CD74 and IP-10 (array 19) were increased over 4-fold in *Fmr1^KO^* males versus females.

Several targets were differentially expressed in the hippocampus as a function of genotype when the mice were fed the AIN-93G/soy diet with IL-30 (array 18) increased 7.0-fold in *Fmr1^KO^* females versus *Fmr1^KO^* males. EMMPRIN (array 18) was increased at least 4.3-fold in *Fmr1^KO^* females compared with all other soy cohorts. FGF4 2.7-fold (array 11) and Tie-2 5.7-fold (array 14) were increased in *Fmr1^KO^* male versus *Fmr1^HET^* female mice; CD4 4.0-fold (array 11) and S100A9 3.7-fold (array 14) increased in *Fmr1^KO^* male versus *Fmr1^KO^* female mice; EpCAM (array 11) increased 2.1-fold in *Fmr1^KO^* males compared with *Fmr1^HET^* females and WT males; NAALADL-1 (array 15) increased 2.8-fold in *Fmr1^KO^* male mice compared with *Fmr1^KO^* females and WT males; FGF6 and FGF10 (array 11) increased at least 4.2-fold in *Fmr1^KO^* males compared with WT males; and FST (array 10) increased at least 14-fold in *Fmr1^KO^* males compared with all other soy cohorts (Table 3).

In terms of the diet-specific differences comparing protein expression for each genotype maintained on an AIN-93G/casein versus AIN-93G/soy diet, 20 proteins were differentially expressed (Table 4). Specifically, PRSS34 and TRANCE were differentially expressed in *Fmr1^HET^* females; BMP-6, IL-1F8, IL-15 Rα and KLB in *Fmr1^KO^* females; CD45 in WT males; and CD4, CD74, CD160, CMG-2, EpCAM, EPHA5, EPHB3, FGF4, FGF6, FZD9, IP-10, MMP-7 and S100A9 in *Fmr1^KO^* males.

### 2.3. Differentially Expressed Proteins in Hypothalamus

Several targets were differentially expressed in the mouse hypothalamus as a function of genotype. When fed AIN-93G/casein, TWEAK R (array 6) was increased 1.7-fold in *Fmr1^KO^* females compared with *Fmr1^KO^* males. When fed AIN-93G/soy, NOV (array 7) was increased 1.2-fold in *Fmr1^HET^* females versus WT males; IL-3 (array 5) increased in *Fmr1^KO^* females versus WT males (null expression); and EGF (array 4) increased 8.1-fold in *Fmr1^KO^* males versus *Fmr1^KO^* females. The diet-specific differences in protein expression within genotypes included 5 proteins, TWEAK R (1.7-fold increase with AIN-93G/casein) and TCK-1 (increased 2.8-fold with AIN-93G/soy) in *Fmr1^KO^* females and EGF (7.2-fold increase with AIN-93G/soy), MMP3 (increased 21-fold with AIN-93G/soy) and PSPN (increased 6.8-fold with AIN-93G/soy) in *Fmr1^KO^* males. Of note, 200 targets were tested in the hypothalamus compared with 640 proteins in the cortex, hippocampus and blood plasma.

### 2.4. Differentially Expressed Proteins in Blood Plasma

Numerous proteins were differentially expressed in the blood plasma as a function of genotype and/or diet in mice. The genotype-specific differences in mice fed AIN-93G/casein included CA2 increased 2.7-fold (array 16) and BID 1.4-fold (array 19) in *Fmr1^HET^* compared with *Fmr1^KO^* females (Table 5). IL-12p40 (null expression in *Fmr1^KO^* mice) (array 5) and EPI (at least 3.9-fold) (array 8) were increased in *Fmr1^HET^* female compared to female and male *Fmr1^K^*^O^ mice. Marapsin was increased 1.6-fold (array 7) in *Fmr1^HET^* females compared with WT males. PSPN was increased 8.1-fold and TRANCE 96-fold (array 8) in *Fmr1^HET^* females compared with *Fmr1^KO^* males. MBL-2 (at least 1.7-fold) (array 7) and VNN1 (at least 3.7-fold) (array 16) were increased in *Fmr1^HET^* females compared to both WT and *Fmr1^KO^* males. SLAM (array 8) was increased at least 4.9-fold in *Fmr1^HET^* female mice compared with all other genotypes fed AIN-93G/casein or AIN-93G/soy. TNF R1 (array 5) was increased 1.5-fold in *Fmr1^KO^* females compared with *Fmr1^HET^* females and WT males. NTB-A (array 16) was increased at least 4.8-fold in *Fmr1^KO^* females compared with *Fmr1^HET^* female and *Fmr1^KO^* male mice. IGFBP-5 increased 2.0-fold, IL-20 (null expression in *Fmr1^KO^* males) (array 4); Nov 2.0-fold (array 7); Siglec-3 3.8-fold, TIM-3 19-fold and uPAR 30-fold (array 13), and BChE 1.6-fold (array 15) in *Fmr1^KO^* female versus *Fmr1^KO^* male mice. DAN increased 1.1-fold (array 7) and ASAM (at least 2.9-fold) (array 12) in *Fmr1^KO^* female versus both WT and *Fmr1^KO^* males. PRL (array 4) was increased at least 3.0-fold in *Fmr1^KO^* females compared with all other genotypes.

Many targets were differentially expressed in blood plasma as a function of genotype in mice fed AIN-93G/soy (Table 5). CXCL16 (array 4) was increased 1.6-fold in *Fmr1^HET^* female mice compared with *Fmr1^KO^* females. RSTN (array 4) was increased at least 2.1-fold in *Fmr1^HET^* female mice compared with WT and *Fmr1^KO^* males and in *Fmr1^KO^* females compared with WT and *Fmr1^KO^* males. VLDL R (array 14) was increased 3.7-fold in *Fmr1^KO^* female versus *Fmr1^HET^* female mice. E-selectin (at least 1.4-fold) (array 4), GAL-3 (at least 1.7-fold) (array 6) and LCN2 (at least 2.2-fold) (array 7) were increased in WT males compared with *Fmr1^HET^* and *Fmr1^KO^* females. RGM-C (array 13) was increased 6.9-fold in *Fmr1^KO^* males compared with *Fmr1^HET^* females. bFGF 1.1-fold (array 5), GAL-3 (at least 1.9-fold) (array 6) and IL-22 Ra1 (at least 12-fold) (array 17) were increased in *Fmr1^KO^* males compared with both *Fmr1^HET^* and *Fmr1^KO^* females. GAL-4 increased 15-fold and EPHB1 29-fold (array 17) in *Fmr1^KO^* male versus *Fmr1^KO^* female mice. SR-A1 (array 13) was increased 4.6-fold in *Fmr1^KO^* male versus WT male mice. JAM-A (at least 1.1-fold) (array 6), SEMA3C (at least 3.5-fold) (array 13) and GDF-9 (at least 5.0-fold) (array 17) were increased in *Fmr1^KO^* males compared with all other genotypes.

In terms of the diet-specific differences comparing protein expression for each genotype maintained on an AIN-93G/casein versus AIN-93G/soy diet, 27 proteins were differentially expressed in blood plasma (Table 6). The differences were specific to a single genotype with ANG, BID, CA-2, EPI, NTN-4, PSPN, SLAM and TRANCE differences in *Fmr1^HET^* females; ASAM, BAFF, BCHE, IL-17F, IL-20, NTB-A, Siglec-3, TIM-3, TNF R1 and VLDL R differences in *Fmr1^KO^* females; no differences in WT males; and bFGF, EPHB1, GAL-4, GDF-9, IL-22 Ra1, JAM-A, SEMA3C, SR-AI and VNN-1 differences in *Fmr1^KO^* males.

### 2.5. Differentially Expressed Proteins in Blood Plasma Comparing Two Diet Studies

Previously, differentially expressed plasma-based proteins were identified in male mice as a function of *Fmr1^KO^* genotype and maintenance on pelleted casein-based infant formula (CIF) versus soy-based infant formula (SIF) diets [12]. The current study compared plasma-based proteins as a function of AIN-93G/casein and AIN-93G/soy as a function of *Fmr1* genotype in both sexes. The major difference between the pelleted infant formula and the purified ingredient diets is that the macronutrient content of infant formula is optimized for human nutrition and contains significantly less protein and more fat than typical mouse diets. The AIN-based diets are formulated for optimal mouse nutrition. In the infant formula diet study, four proteins were differentially expressed in blood plasma in males as a function of WT versus *Fmr1^KO^* genotype including B7-1 (1.75-fold) with SIF and ICAM-1 (3.71-fold), MFG-E8 (4.77-fold) and PRL (0.34-fold) with CIF [12]. In the current study testing AIN diets, there were no statistically significant differences in these proteins as a function of *Fmr1* genotype in male mice. To increase statistical power, the analysis of the AIN-93G diet data for these targets was repeated considering only males and using the unpaired *t*-test with Welch’s correction. There were still no statistically significant differences dependent on genotype in males. PRL remains a target of interest because it was increased in blood plasma from *Fmr1^KO^* females compared with all other genotypes when mice were fed AIN-93G/casein (Figure 1), indicating sex- and *Fmr1* genotype-specific regulation.

In terms of diet, eight proteins were differentially expressed as a function of CIF or SIF in the infant formula study (Table 7). Less stringent *t*-test statistics with Welch’s corrections indicated that only IGFBP-5 was significantly different in both the infant formula and AIN diet studies with an increase of 1.4–1.6-fold in *Fmr1^KO^* mice in response to the soy-based diet (Table 7, Figure 2). Of interest, there were also statistically significant differences in IGFBP-5 for *Fmr1^HET^* and *Fmr1^KO^* females, albeit in opposite directions suggesting that this protein is regulated as a function of sex, genotype and diet.

### 2.6. Differentially Expressed Top Proteins as a Function of Fmr1 Genotype and AIN-93G Diets

We found 202 significant differences with *p* values ranging from 0.0007 to 0.0496 (Appendix A). Targets were from the cortex (*n* = 47), hippocampus (*n*= 54), hypothalamus (*n* = 9) and plasma (*n* = 92). The majority of significantly different comparisons as a function of AIN-93G/casein versus AIN-93G/soy diets within individual genotypes were with *Fmr1* mutant mice [*Fmr1^HET^* females (*n* = 15), *Fmr1^KO^* females (*n* = 23), WT males (*n* = 6) and *Fmr1^KO^* males (*n* = 26)]. There were twice as many differences in females (*Fmr1^HET^* versus *Fmr1^KO^*) compared with males (WT versus *Fmr1^KO^*) with the AIN-93G/casein diet and 2.5-fold more differences in males compared with females with the AIN-93G/soy diet. Only five proteins were differentially expressed in two tissues including BID (cortex and plasma), EGF (hippocampus and hypothalamus), NOV (hypothalamus and plasma), PSPN (hypothalamus and plasma) and TRANCE (hippocampus and plasma). TRANCE was the only protein with differential expression among similar comparisons when comparing tissues, i.e., *Fmr1^HET^* females fed AIN-93G/casein versus AIN-93G/soy as well as *Fmr1^HET^* females versus *Fmr1^KO^* males fed AIN-93G/casein.

None of the targets passed a conservative Bonferroni’s multiple hypothesis correction with a significance threshold of *p* < 0.00002 (based on dividing *p* = 0.05 by 2120 comparisons from 640 targets for the cortex, hippocampus and plasma plus 200 targets for the hypothalamus). Using the Benjamini–Hochberg’s test, which is useful when there is a large number of comparisons and provides a moderate reduction in false positives, 27 statistically significant differences were generated among 19 proteins (Table 8). There was no overlap between tissues. One protein was differentially expressed in the cortex (CD14), four proteins in the hippocampus (CD4, IL-15 Ra, KLB and PRSS34), one protein in the hypothalamus (NOV), and thirteen proteins in plasma (ASAM, EPI, GAL-3, GDF-9, JAM-A, LCN-2, MBL-2, PRL, RSTN, SEMA3C, SLAM, SR-A1 and VNN1). Of note, GDF-9 was the only protein differentially expressed between WT and *Fmr1^KO^* male mice (Figure 3). PRL was the top significantly different target (*Fmr1^KO^* female versus *Fmr1^KO^* males fed AIN-93G/casein, *p* = 0.0007) (Figure 1). STRING analysis indicates protein association networks (Figure 4).

### 2.7. Differentially Expressed Proteins in Hippocampus and Hypothalamus Comparing Mass Spectrometry and Quantibody Arrays

Protein expression in the hippocampus and hypothalamus of male WT and *Fmr1^KO^* littermate mice fed AIN-93G/casein versus AIN-93G/soy diets was previously quantitated by global shotgun proteomics [14]. Merging the shotgun proteomics and Quantibody datasets identified overlap in the expression of 33 proteins. One protein was expressed in both the hippocampus and hypothalamus in both studies, i.e., CLU. Thirty-one proteins were expressed in the hippocampus in both studies including C1QBP, CA2, CA4, CADM3, CAMK4, CD34, CD47, CD200, CHL1, CNTN1, CNTN2, CRELD1, EPHA4, FDPS, GAPDH, ICAM-5, IGSF8, JAM-C, LRPAP, LTA4H, MANF, MOG, NCAM-1, OLFM-1, PPM1A, PREP, PSMB6, SIRPA, THOP1, TSC22D1 and UCHL1. One protein was expressed in the hypothalamus in both studies, i.e., GAL-1. None of these common targets were differentially expressed in the respective hippocampal and hypothalamic tissue by Quantibody arrays. More than half exhibited differential expression by mass spectrometry where ten proteins exhibited both diet- and genotype-specific differences in the hippocampus, i.e., CADM3, CD200, CNTN1, EPHA4, IGSF8, LTA4H, MOG, NCAM-1, SIRPA and THOP1 (Figure 5). Other differences included a reduced expression of GAL-4 in *Fmr1^KO^* compared with WT hypothalamus with both AIN-93G/casein and AIN-93G/soy diets; a reduced expression of CNTN2 hippocampus in *Fmr1^KO^* compared with WT mice in response to an AIN-93G/casein diet; an increased expression of CA2 and FDPS in hippocampus in WT compared with *Fmr1^KO^* mice in response to an AIN-93G/casein diet; a reduced expression of CLU and an increased expression of UCHL1 in the hippocampus in *Fmr1^KO^* compared with WT with an AIN-93G/soy diet; and a reduced expression of C1QBP and CAMK4 in *Fmr1^KO^* hippocampus in response to an AIN-93G/soy versus AIN-93G/casein diet (Appendix A).

## 3. Discussion

We hypothesize that consuming soy protein-rich diets during postnatal development is an environmental exposure that increases the risk of seizures, autism and obesity, particularly in vulnerable populations such as those with FXS. Here, we screened for brain- and blood-based protein biomarkers that were dysregulated in response to casein and soy protein isolate (SPI)-matched purified ingredient diets as a function of *Fmr1* genotype and sex. We analyzed three brain regions, the cortex, hippocampus and hypothalamus, as well as blood plasma. The cortex is responsible for higher-level cognitive function including perception, attention, language and consciousness. Both the hippocampus and hypothalamus are part of the limbic system. The hippocampus is important for memory formation and consolidation. The hypothalamus produces hormones that regulate body temperature, heart rate, blood pressure, appetite, mood, sleep, muscle and bone growth, and the release of hormones from other glands. Blood plasma is an easily accessible, less invasive, cost-effective biomarker compared with cerebral spinal fluid and brain imaging. We identified several proteins that were differentially expressed as a function of *Fmr1* genotype and/or AIN-93G/soy diet including PRL, IGFBP5 and GDF-9. No proteins under study were consistently altered across tissue type as a function of *Fmr1* genotype or diet. No proteins under study were differentially regulated in both global shotgun proteomics and Quantibody arrays analyses. It would be beneficial to the FXS field to identify blood-based biomarkers that correlate with brain protein levels and that are responsive to FMRP, disease severity and therapeutic intervention. This study investigates the effect of an extrinsic variable (soy diet) on protein expression as a function of *Fmr1* genotype, sex and tissue.

We find *Fmr1* genotype-, sex-, diet- and tissue-specific differences in biomarker expression in mice. Of interest, more proteins were differentially expressed in response to AIN-93G/casein compared with AIN-93G/soy, particularly in the hippocampus followed by the cortex and blood plasma. Of the 25 proteins identified in blood plasma from *Fmr1^HET^* or *Fmr1^KO^* female mice, 88% were differentially expressed in response to AIN-93G/casein whereas the 16 differentially expressed proteins identified in WT or *Fmr1^KO^* male plasma were all in response to AIN-93G/soy. These findings are important because the majority of preclinical studies are conducted in male mice, particularly when studying models of X-chromosome-linked disorders such as FXS, and because a large number of vivariums utilize grain- and/or soy-based chows for routine animal husbandry.

We found no overlap in the differential expression of Quantibody targets between the tissue types tested. We also found no overlap in differentially expressed proteins when merging the Quantibody data with a prior global shotgun proteomics study to identify biomarkers in young adult *Fmr1^KO^* mice in response to AIN-93G/soy [14]. Mass spectrometry is considered a more sensitive method with high specificity for quantitating protein levels versus ELISA-based technology; however, mass spectrometry is more expensive and requires highly specialized equipment and more complex sample preparation.

We previously analyzed the effects of CIF and SIF on weight gain, neurodevelopment and blood plasma biomarkers in *Fmr1^KO^* mice [12]. Infant formula is optimized for human development and contains significantly less protein and more fat than typical mouse diets. The consumption of SIF was associated with the altered expression of numerous plasma proteins including elevated LEP and NEP in comparison with both CIF and Teklad 2019 chow in WT mice and elevated NEP in *Fmr1^KO^* mice compared with Teklad 2019. The altered macronutrient content of the SIF in conjunction with soy protein likely underlies these findings as we did not observe the altered expression of these protein with casein and soy-based AIN-93G diets. Of interest, PRL expression in plasma was significantly elevated 2.9-fold in *Fmr1^KO^* compared with WT male mice when fed CIF as well as 3.6-fold higher in *Fmr1^KO^* male mice fed CIF versus Teklad 2019 chow. With AIN-93 diets, PRL levels were significantly elevated in *Fmr1^KO^* female mice as a function of genotype and sex selectively when fed AIN-93G/casein. Thus, PRL appears to be an *Fmr1* genotype-responsive biomarker in the context of casein protein.

PRL is a hormone that functions in milk production, mammary gland development, testicular function, copulatory behavior, immune response, osmoregulation and angiogenesis [20,21]. Hyperprolactinemia can disrupt the menstrual cycle; cause infertility, galactorrhea (nipple discharge) and erectile dysfunction; and is associated with seminal fluid abnormalities, arrest of spermatogenesis and impaired sperm motility [22]. PRL is found in breast milk and correlates with plasma levels [23]. Elevated PRL in *Fmr1^KO^* female mice could affect pup development; for example, PRL levels are associated with maternal behavior and dam response to nursing pups [24,25]. There is a report of elevated baseline PRL levels in males with FXS, which was not confirmed by another study [26,27]. Male *Fmr1^KO^* mice exhibit an increased proliferation of Sertoli cells from embryonic day 12 to postnatal day 15 [28], and Sertoli cells are potential targets for PRL action in testis [29]. Testicular function in men with FXS is not mediated by hormones (LH, FSH, testosterone, dihydrotestosterone, androstenedione, 17-hydroxyprogesterone, progesterone, dehydroepiandrosterone and dehydroepiandrosterone-sulfate) [30], but considering the dearth of literature regarding hormone levels in FXS patients and the strong macroorchidism phenotype, the dysregulation of PRL may be a viable biomarker and therapeutic target.

Plasma IGFBP5 was the only differentially expressed Quantibody target as a function of soy in both the pelleted infant formula and matched AIN-93G diet studies. While there is no literature to our knowledge regarding a role of IGFBP5 in FXS, insulin signaling underlies circadian and cognitive deficits in FXS flies and mice [31,32,33]. High-dose genistein treatment is associated with increased IGFBP5 expression and the activation of apoptosis and protein tyrosine kinase signaling pathways in broiler hens [34]. Soy is rich in phytoestrogens including genistein and daidzein. IGFBP family members including IGFBP5 are inversely associated with autism severity [35], and are potential peripheral biomarkers for numerous psychiatric disorders [36].

Plasma GDF-9 from mice fed AIN-93G/soy was the only Quantibody target that was differentially expressed between WT and *Fmr1^KO^* male mice after a Benjamini–Hochberg’s statistical correction. GDF-9 is a member of the TFG-β superfamily that functions in follicle maturation with genetic variants found in premature ovarian insufficiency (POI) [37,38,39]. POI is also associated with the *Fmr1* premutation. In males, GDF-9 is expressed in large spermatocytes and the round spermatids of the testes and inhibits membrane localization of tight junction proteins claudin-11, occludin and zonula occludens-1 [39]. GDF-9 disrupts tight junction integrity and affects acrosome integrity rate, sperm concentration and sperm positive rheotaxis [40,41,42].

A STRING analysis for protein interactions found no physical interactions between the 19 proteins identified by Quantibody arrays; however, several functional associations were found that centered on CD4. CD4 was differentially expressed in the hippocampus as a function of sex (4-fold increase in *Fmr1^KO^* males versus females) and diet (9.5-fold increase in *Fmr1^KO^* males fed AIN-93G/soy versus AIN-93G/casein). Potential confounding issues including the use of small cohorts and not assessing and controlling for the estrous cycle in females are not issues with this target as the highest variability between mice was in the *Fmr1^KO^* male cohort with an *n* = 8. CD4 is found on the surface of helper T immune cells, which play a vital role in fighting infections. Persons with FXS tend to have hypoactivation of the immune system, for example, increased otitis media (ear infections).

The implications of these findings are manifold. First, several proteins were differentially expressed as a function of sex and diet. Considering that the majority of *Fmr1^KO^* mouse studies only test male mice and that most publications do not report diet, it is not surprising that there is not a consensus blood-based biomarker panel for FXS. The lack of reporting of routine mouse husbandry details including diet and bedding is concerning. Second, there was no overlap in differentially expressed Quantibody targets between brain tissue and plasma indicating that expression in the brain is not reflected in the blood under our test conditions. Third, the identified biomarkers such as PRL are hormones that are affected by diet, which has implications for nursing and nurturing during postnatal development. The selective effect of casein versus soy on PRL levels and in *Fmr1^KO^* but not *Fmr1^HET^* females suggests that different breeding schemes and maternal *Fmr1* genotype and PRL levels will affect outcomes in progeny. Indeed, milk-borne bioactive factors transmit signals from dams to pups during nursing and affect offspring social dominance [43], while reduced maternal FMRP levels in dams causes hyperactivity and altered social behavior in wild type offspring [44,45,46].

Potentially confounding issues include the female estrous cycle and fasting status. We did not measure estrous state in female mice. The mice were 4 months old at tissue harvest and mouse cyclicity in C57BL/6J ceases at 13–16 months of age [47]. For females, we compared *Fmr1^HET^* and *Fmr1^KO^* littermates. It is not possible to generate WT and *Fmr1^KO^* female littermates because the *Fmr1* gene is located on the X-chromosome. The blood and tissue samples were collected after 4 h fasting in this study. Samples were collected during non-fasting in our prior study [12]. Fasting can decrease some proteins in blood plasma and increase others while testing under non-fasting conditions can be confounded by intermittent eating by the mice. The use of anesthesia and blood anticoagulants can also affect protein expression [48,49]. For animal experiments, a minimum of 3 independent biological replicates is recommended and we used *n* = 4–11 mice per cohort. We acknowledge that interesting findings need to be replicated with larger study cohorts and by multiple methods. We did not exclude potential outliers here due to the smaller study size and because these data points may represent genuine, natural variation within the population. If a study is underpowered, the risk of Type II errors (false negatives) is increased. To adequately power a study with eight cohorts (four genotypes, two diets) and analyze by ANOVA with a power of 0.8, a medium size effect (Cohen’s *f* = 0.25) and a significance level of 0.05 would require over 200 mice per cohort, which is not feasible, violates The 3Rs Principle, and would not be approved by the local IACUC.

Future research directions include the study of the effects of diet–drug and gut–brain interactions on protein expression in *Fmr1^KO^* mice. Mice are the major model for preclinical drug testing and diet–drug interactions could impede moving the most effective drugs into clinical trials. The identification of biomarkers that respond to genotype, diet and drug interventions is expected to facilitate drug discovery and the monitoring of therapeutic efficacy. Understanding how the gut communicates with the brain in conjunction with identifying diet-derived bioactive components affecting those communications is also expected to facilitate drug discovery for FXS. In addition, future research includes mechanistic studies to understand how bioactive components in the diet, for example, soy phytoestrogens mediate the effects on *Fmr1^KO^* outcomes. There are numerous molecular signaling pathways implicated in FXS with metabotropic glutamate receptor 5 (mGluR_5_) being the most studied. mGluR_5_ colocalizes with estrogen receptors (ER) at synapses suggesting that soy phytoestrogens could mediate effects by interfering with mGluR_5_/ER signaling [18].

## 4. Materials and Methods

### 4.1. Mouse Husbandry

Mice were obtained from our colony maintained for over 20 years at the University of Wisconsin, Madison, WI USA; the *Fmr1^KO^* strain in the C57BL/6J background was originally provided by Dr. William Greenough’s laboratory at the University of Illinois, Urbana-Champaign, IL, USA. *Fmr1^HE^*^T^ females and *Fmr1^KO^* males used for breeding were transferred to test diets at least 12 days prior to mating. Breeding pairs were housed in microisolator cages with Shepherd’s Cob + Plus, ¼-inch cob bedding on a 12 h (0600–1800) light cycle with ad libitum access to food and dechlorinated water (specific diets described below). Due to the X-chromosome location of the *Fmr1* gene, the breeding scheme generated *Fmr1^HET^* and *Fmr1^KO^* females and WT and *Fmr1^K^*^O^ males. WT and *Fmr1^K^*^O^ female littermates cannot be generated. Genotypes of offspring were determined by PCR analysis of DNA extracted from tail biopsies taken at weaning as described [50], or by Transnetyx automated genotyping service (Memphis, TN, USA). The animal study protocol was approved by the Institutional Animal Care and Use Committee (IACUC) at the University of Wisconsin, Madison (protocol code M005224).

### 4.2. Diets

The test diets were previously described [15]. Briefly, purified ingredient diets formulated by Envigo included casein-based TD.180374, which is a modification of AIN-93G (Envigo TD.94045) albeit with increased sodium at 2 g/kg diet (0.2%) to match the sodium content of soy protein isolate-based TD.180375. TD.180374 contains 17.7% protein by percent weight, 59.8% carbohydrate and 7.2% fat corresponding to 18.9%, 63.8% and 17.3%, respectively, % kcal with a total energy density of 3.7 kcal/g. Red food dye was added for visual differentiation. TD.180375 is modified from AIN-93G to replace casein with soy protein isolate and to match micronutrients to the control diet TD.180374 including 0.5% calcium, 0.3% available phosphorus, 0.2% sodium, 0.36% potassium, 0.3% chloride and 0.05% magnesium. TD.180375 contains 17.8% protein by % weight, 60.7% carbohydrate and 7.2% fat corresponding to 18.8%, 64.1% and 17.1%, respectively, % kcal with a total energy density of 3.8 kcal/g. Green food dye was added for visual differentiation. Upon delivery, diets were transferred to sealed Tupperware containers and stored at 4 °C prior to use. Complete formulations are provided in Appendix A.

### 4.3. Study Design and Tissue Collection

The study design was a 2 × 2 factorial design with two main effects and interactions: diet (casein and soy) and genotype (female *Fmr1^HET^* and *Fmr1^KO^*, male WT and *Fmr1^KO^*). Mice were maintained on their respective diets for 4 months. Blood and tissue samples were collected during the light phase after at least 4 h fasting [15]. Mice were anesthetized with isoflurane for 90 s (mid-light cycle, 1300–1530). Blood samples were collected from the inferior vena cava with a 21-gauge butterfly needle. Exsanguination euthanized the mouse. The blood was mixed with a sodium heparin anticoagulant, and after all blood samples were collected, spun at 5000 rpm for 10 min, followed by the plasma layer, quick frozen and stored at −80 °C. The left and right cortex, hippocampus and hypothalamus were immediately dissected from the brain after blood collection, flash frozen on dry ice, and stored at −80 °C. For the dissections, the hypothalamus, which was the smallest tissue collected, was dissected first. The whole brain was placed ventral side up by flipping over in the skull and jeweler’s forceps were used to scoop out the translucent-colored, spherical-shaped hypothalamus. There was a depression where the hypothalamus was excised. The brain was then flipped over in the skull dorsal side up. The hippocampus was dissected by using a razor blade to scrape up the lower part of the left and right cortex followed by excision of the hippocampus with jeweler’s forceps. The left and right cortex were dissected last. The banked blood plasma (50 μL), left cortex, hippocampus and hypothalmus tissue from 62 mice were shipped overnight on dry ice to RayBiotech, Inc. and used to probe mouse protein arrays (RayBiotech, Inc., Peachtree Corners, GA, USA).

The study adhered to ARRIVE guidelines. Female *Fmr1^HET^* mice were randomly assigned to test diets for breeding purposes. Mice were maintained on matched casein- and soy protein isolate-based purified ingredient diets containing AIN-93G-recommended levels of macro- and micro-nutrients. The experimental unit was a single animal. Mice from a minimum of three litters were included in each cohort. The number of mice per cohort were *Fmr1^HET^* female/AIN-93G/casein (*n* = 5), *Fmr1^KO^* female/AIN-93G/casein (*n* = 8), WT male/AIN-93G/casein (*n* = 4), *Fmr1^KO^* male/AIN-93G/casein (*n* = 9), *Fmr1^HET^* female/AIN-93G/soy (*n* = 9), *Fmr1^KO^* female/AIN-93G/soy (*n* = 8), WT male/AIN-93G/soy (*n* = 11) and *Fmr1^KO^* male/AIN-93G/soy (*n* = 8). The P.I. and staff were aware of diet group allocation throughout the study as diets were color coded pink (AIN-93G/casein) and green (AIN-93G/soy) to avoid mix up of treatments over a 4-month dosing period. Scientists conducting the microarray analyses were blind to genotype and diet identities.

### 4.4. Quantibody Cytokine Arrays

Samples were analyzed by RayBiotech’s Quantibody Mouse Full Testing Service, which included sample preparation, BCA protein determination and array runs for 248 samples (62 mice, 4 tissues per mouse). Plasma, cortex and hippocampus were run on the Quantibody^®^ Mouse Cytokine Antibody Array 640 (QAM-CAA-640), which is a combination of 16 non-overlapping arrays to quantitatively measure 640 mouse cytokines. Arrays include QAM-CYT-4, QAM-CYT-5, QAM-CYT-6, QAM-CYT-7, QAM-CYT-8, QAM-CYT-9, QAM-CYT-10, QAM-CYT-11, QAM-CYT-12, QAM-CYT-13, QAM-CYT-14, QAM-CYT-15, QAM-CYT-16, QAM-CYT-17, QAM-CYT-18 and QAM-CYT-19. The protein concentrations were approximately 80 μg/mL for plasma, hippocampus and cortex, and 1 μg/mL was loaded per array. Hypothalamus samples ranged from 4 to 12 μg/mL and there was not enough material for the 640 arrays. Thus, hypothalamus samples were run on the Quantibody^®^ Mouse Cytokine Antibody Array 4000 (QAM-CAA-4000), which is a combination of 5 non-overlapping arrays to quantitatively measure 200 mouse cytokines. Arrays include QAM-CYT-4, QAM-CYT-5, QAM-CYT-6, QAM-CYT-7 and QAM-CYT-8. All samples were analyzed at 1 μg/mL.

Quantibody^®^ arrays are a multiplexed sandwich ELISA platform that allow the simultaneous quantification of multiple proteins [51]. The advantage includes combining the high detection sensitivity and specificity of an ELISA with the high throughput of an array. The technology utilizes a pair of protein-specific antibodies for detection. First, a capture antibody is bound to a glass surface. The sample is incubated with the capture antibody trapping the target protein on a solid surface. Then a second biotin-labeled detection antibody that recognizes a different epitope of the target protein is added. The protein–antibody–biotin complex is visualized by the addition of streptavidin-conjugated Cy3 equivalent dye and a laser scanner. The array format on a glass support allows the detection of multiple proteins in one experiment. The glass slide is divided into 16 wells with identical antibody arrays. Positive controls and antibodies are arrayed in quadruplicate. A tray holds four slides allowing for an automated robotic high-throughput process of 64 arrays at a time. Standard curves are generated for each protein and used to calculate the concentration of unknown samples using Quantibody^®^ Q-Analyzer software version 8.40.4, which is an Excel-based program for data analysis. Specifically, the Q-Analyzer Tool (catalog QAM-CAA-640-SW) determines protein concentration by a copy and paste process, automatically marks outlier spots, allows for intra- and inter-slide normalization for large sample sets, and uses two positive controls for normalization within each array. Result output is presented in pg/mL. The complete service reports are provided as Appendix A.

### 4.5. Data Analysis

Statistical significance was determined by a 2-way ANOVA with Tukey’s multiple comparison test using GraphPad Prism for macOS version 10.4.1 (532). Two methods were used to correct for multiple comparisons to reduce the risk of false positives (Type 1 errors), the Bonferroni correction and the Benjamini–Hochberg test.

### 4.6. STRING Protein Interaction Analysis

STRING is a database of known and predicted protein–protein interactions including direct (physical) and indirect (functional) associations derived from five main sources: genomic context predictions, high-throughput lab experiments, conserved co-expression, automated text mining and previous knowledge in databases. The STRING database currently covers 59,309,604 proteins and 12,535 organisms [52]. The 19 proteins that passed the Benjamini–Hochberg correction after Quantibody cytokine array analysis were analyzed using the STRING web resource STRING Version 12.0 for known and predicted protein–protein interactions in the full string network (functional and physical) protein interactions.

## 5. Conclusions

In conclusion, several protein biomarkers were identified that were differentially expressed as a function of *Fmr1* genotype and/or AIN-93G/soy diet. The proteins of significant disease relevance included PRL, IGFBP5 and GDF-9. Of note, no proteins were identified that were consistently altered across tissue type as a function of *Fmr1* genotype or diet. No proteins were identified that were differentially regulated as quantitated by both global shotgun proteomics and Quantibody arrays.

## Figures and Tables

**Figure 1 ijms-26-06137-f001:**
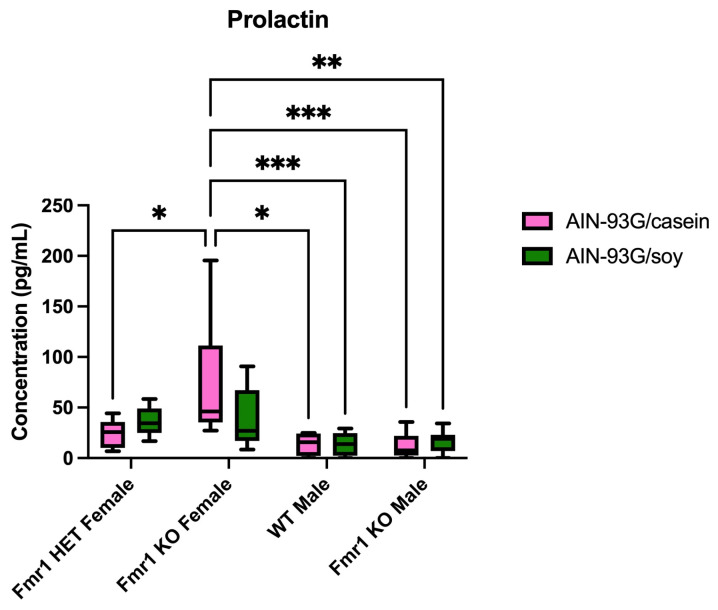
Prolactin (PRL) expression in blood plasma as function of *Fmr1* genotype and AIN-93G diet. Mice on AIN-93G/casein included *n* = 5 *Fmr1^HET^* female, *n* = 8 *Fmr1^KO^* female, *n* = 4 WT male and *n* = 9 *Fmr1^KO^* male. Mice on AIN-93G/soy included *n* = 9 *Fmr1^HET^* female, *n* = 8 *Fmr1^KO^* female, *n* = 11 WT male and *n* = 8 *Fmr1^KO^* male. The average concentration in blood plasma in pg/mL was plotted versus genotype. Statistics were determined by 2-way ANOVA and Tukey’s multiple comparison tests denoted by *p* < 0.05 (*), *p* < 0.01 (**) and *p* < 0.001 (***). ANOVA results: interaction F (3, 54) = 2.055, *p* = 0.1170; genotype F (3, 54) = 8.379, *p* = 0.0001; and diet F (1, 54) = 0.3681, *p* = 0.5466.

**Figure 2 ijms-26-06137-f002:**
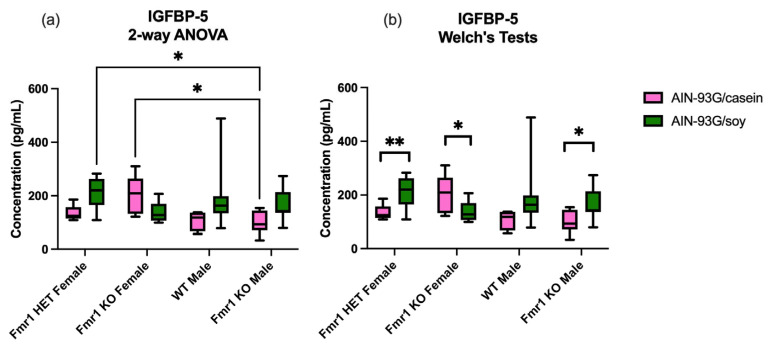
IGFBP-5 expression in blood plasma as function of *Fmr1* genotype and AIN-93G diet. Mice on AIN-93G/casein included *n* = 5 *Fmr1^HET^* female, *n* = 8 *Fmr1^KO^* female, *n* = 4 WT male and *n* = 9 *Fmr1^KO^* male. Mice on AIN-93G/soy included *n* = 9 *Fmr1^HET^* female, *n* = 8 *Fmr1^KO^* female, *n* = 11 WT male and *n* = 8 *Fmr1^KO^* male. The average concentration in plasma in pg/mL was plotted versus genotype. (**a**) Statistics were determined by 2-way ANOVA and Tukey’s multiple comparison tests denoted by *p* < 0.05 (*). ANOVA results: interaction F (3, 54) = 4.203, *p* = 0.0096; genotype F (3, 54) = 1.268, *p* = 0.2946; and diet F (1, 54) = 4.178, *p* = 0.0458; (**b**) Statistics were determined by unpaired *t*-tests with Welch’s corrections for each genotype where significant differences are denoted by *p* < 0.05 (*) and *p* < 0.01 (**).

**Figure 3 ijms-26-06137-f003:**
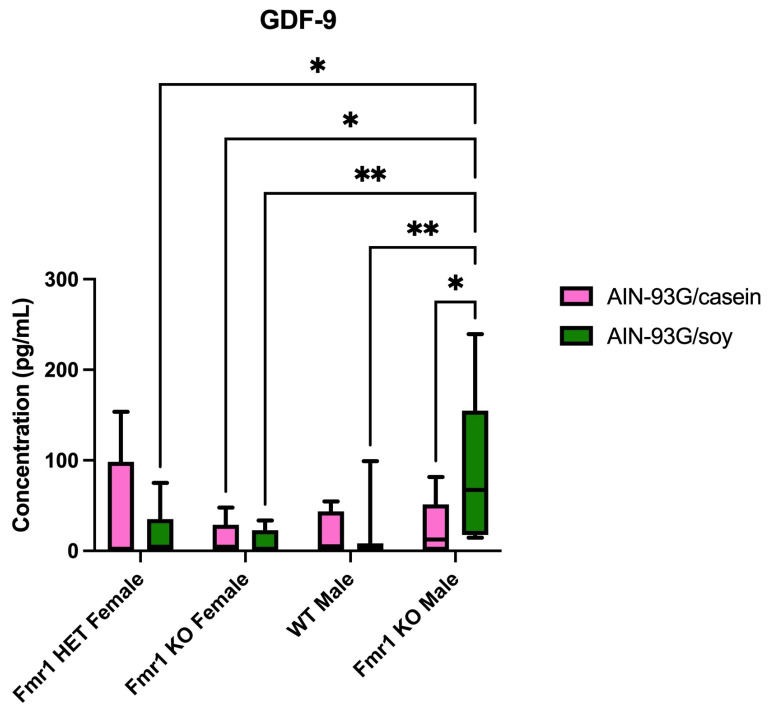
GDF-9 expression in blood plasma as a function of *Fmr1* genotype and AIN-93G diets. Mice on AIN-93G/casein included *n* = 5 *Fmr1^HET^* female, *n* = 8 *Fmr1^KO^* female, *n* = 4 WT male and *n* = 9 *Fmr1^KO^* male. Mice on AIN-93G/soy included *n* = 9 *Fmr1^HET^* female, *n* = 8 *Fmr1^KO^* female, *n* = 11 WT male and *n* = 8 *Fmr1^KO^* male. The average concentration in plasma in pg/mL was plotted versus genotype. Statistics were determined by 2-way ANOVA and Tukey’s multiple comparison tests denoted by *p* < 0.05 (*) and *p* < 0.01 (**). ANOVA results: interaction F (3, 54) = 3.322, *p* = 0.0264; genotype F (3, 54) = 4.170, *p* = 0.0100; and diet F (1, 54) = 0.5326, *p* = 0.4687.

**Figure 4 ijms-26-06137-f004:**
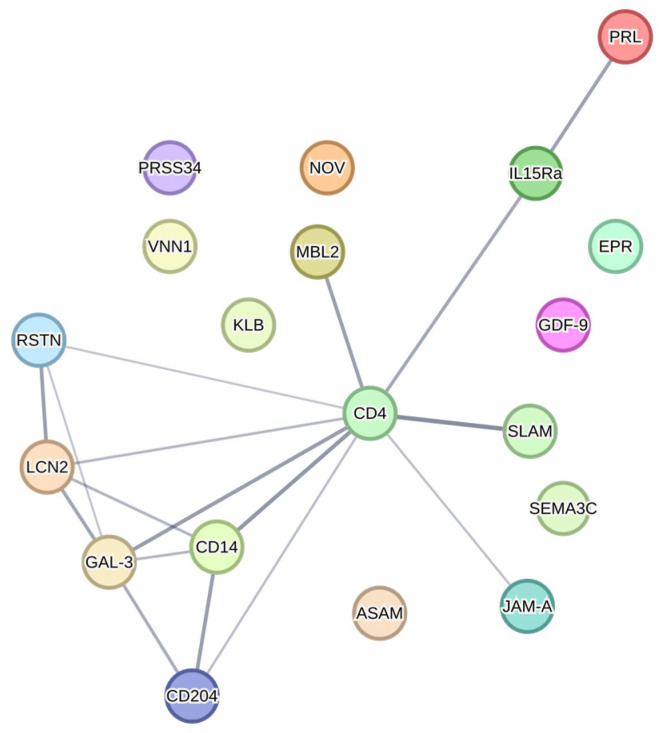
Functional protein associations between top protein targets after Benjamini–Hochberg’s correction. Functional connections among 19 proteins that passed statistical significance after Benjamini–Hochberg’s correction were analyzed by STRING analysis (STRING version 12.0). Proteins included ASAM, CD4, CD14, EPI (also known as EPR), GAL-3, GDF-9, IL-15 Ra, JAM-A, KLB, LCN-2, MBL-2, NOV, PRL, PRSS34, RSTN, SEMA3C, SLAM, SR-A1 (also known as CD204) and VNN1. No interactions were found with the physical subnetwork that would have indicated proteins were part of a physical complex. Functional connections among the proteins are displayed with darker lines indicating stronger associations between proteins.

**Figure 5 ijms-26-06137-f005:**
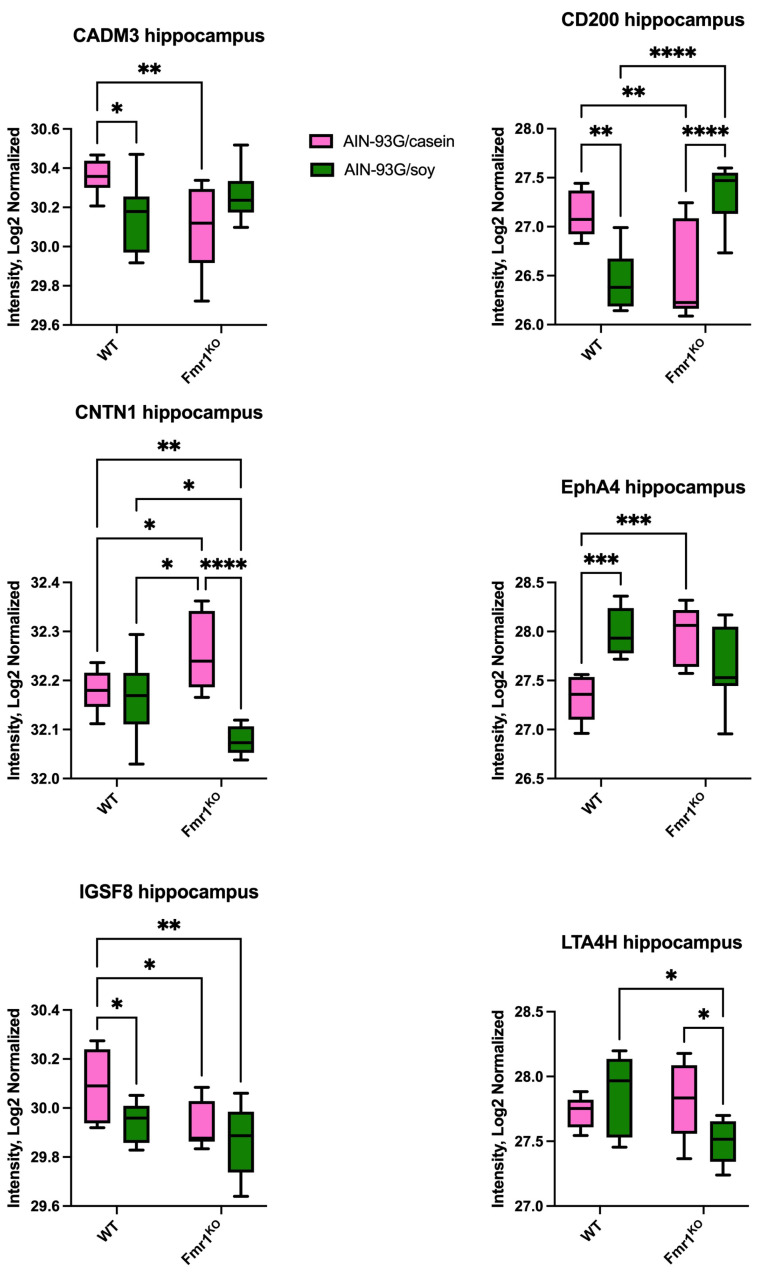
Protein expression by mass spectrometry as a function of *Fmr1* genotype and AIN-93G diet for select targets overlapping with Quantibody arrays. Data included 3 biological and 3 technical replicates for each cohort, which included WT and *Fmr1^KO^* male mice on AIN-93-G/casein and AIN-93G/soy diets [14]. The average intensity, log2 normalized in the hippocampus was plotted versus genotype/diet cohort. Statistics were determined by 2-way ANOVA and Tukey’s multiple comparison tests where significant differences are denoted by *p* < 0.05 (*), *p* < 0.01 (**) and *p* < 0.001 (***).

**Table 1 ijms-26-06137-t001:** Differential protein expression in mouse cortex as a function of genotype.

Diet ^a^	*Fmr1^HET^* Female ^b^	*Fmr1^KO^* Female ^b^	WT Male ^b^	*Fmr1^KO^* Male ^b^
AIN-93G/casein	CCL6 FCRL5 OLFM-1	CT-1 Eotaxin IGFBP-2 KC LEP MIG OPG SCF	BAMBI BID BMP-7 EPHA1 FAM3C	CD99-L2
AIN-93G/soy	IL-1β	PODXL		CD164 CST3

^a^ Raw data is available in the Appendix A. ^b^ Protein names and functions are listed in Appendix A. Red color indicates increased expression and blue color indicates decreased expression.

**Table 2 ijms-26-06137-t002:** Differential protein expression in mouse cortex as a function of diet.

Array ^a^	Protein ^b^	*Fmr1^HET^* Female	*Fmr1^KO^* Female	WT Male	*Fmr1^KO^* Male
4	OPG		↑ 9-fold casein		
	SCF		↑ 2.4-fold casein		
5	Eotaxin		↑ 3.2-fold casein		
	LEP		↑ 38-fold casein		
	MIG		↑ 63-fold casein		
6	CT-1		↑ 2.7-fold casein		
	IL-3 Rb		↑ 22-fold casein		
9	B7-2	↑ 4.1-fold soy			
10	CD14	↑ 11-fold casein			
12	OLFM-1	↓ 1.6-fold casein			
14	CD164				↑ 2.4-fold soy
15	CAMK4	↓ 6.9-fold casein			
17	GHR			↑ 41-fold casein	
18	FCRL5	↑ 49-fold casein			
19	BID			↑ 26-fold casein	
	BMP-7			↑ 10-fold casein	
	EPHA1			↑ 11-fold casein	
	FAM3C			↑ 11-fold casein	

^a^ Raw data is available in the Appendix A. ^b^ Protein names and functions are listed in Appendix A. Casein refers to AIN-93G/casein and soy refers to AIN-93G/soy diet. ↑ refers to an increase in fold change. ↓ refers to a decrease in fold change.

**Table 3 ijms-26-06137-t003:** Differential protein expression in mouse hippocampus as a function of genotype.

Diet ^a^	*Fmr1^HET^* Female ^b^	*Fmr1^KO^* Female ^b^	WT Male ^b^	*Fmr1^KO^* Male ^b^
AIN-93G/casein	CHST4 MMP-10 PRSS34 SPB10 TRANCE	IL-20 Rb IL-15 Ra LIF KLB	CD45 EGF	CD74 IP-10
AIN-93G/soy		IL-30 EMMPRIN		CD4 EpCAM FGF-4 FGF-6 FGF-10 FST NAALADL-1 S1000A9 Tie-2

^a^ Raw data is available in the Appendix A. ^b^ Protein names and functions are listed in Appendix A. Red color indicates increased expression and blue color indicates decreased expression.

**Table 4 ijms-26-06137-t004:** Differential protein expression in mouse hippocampus as a function of diet.

Array ^a^	Protein ^b^	*Fmr1^HET^* Female	*Fmr1^KO^* Female	WT Male	*Fmr1^KO^* Male
Q8	TRANCE	↑ 43-fold casein			
Q9	CD45			↑ 1.9-fold casein	
Q10	IL-1F8		↑ 7.9-fold casein		
	IL-15 Ra		↑ 5.0-fold casein		
	KLB		↑ 38-fold casein		
Q11	CD4 EpCAM EPHA5 FGF4 FGF6				↑ 9.5-fold soy ↑ 2.3-fold soy ↑ 5.4-fold soy ↑ 3.3-fold soy ↑ *-fold soy
Q12	PRSS34	↑ *-fold casein			
Q14	S100A9				↑ 3.9-fold soy
Q16	BMP-6		↑ 3.7-fold soy		
Q19	CD74 CD160 CMG-2				↑ 6.2-fold casein ↑ 17.3-fold casein ↑ 2.9-fold casein
	EPHB3				↑ 23.9-fold casein
	FZD9				↑ 10.3-fold casein
	IP-10				↑ 4.4-fold casein
	MMP-7				↑ 228-fold casein

^a^ Raw data is available in the Appendix A. ^b^ Protein names and functions are listed in Appendix A. * Fold change cannot be calculated as one group has a mean of 0 pg/mL. Casein refers to AIN-93G/casein and soy refers to AIN-93G/soy diet. ↑ refers to an increase in fold change.

**Table 5 ijms-26-06137-t005:** Differential protein expression in mouse blood plasma as a function of genotype.

Diet ^a^	*Fmr1^HET^* Female ^b^	*Fmr1^KO^* Female ^b^	WT Male ^b^	*Fmr1^KO^* Male ^b^
AIN-93G/casein	BID CA2 EPI IL-12p40 Marapsin MBL-2 PSPN SLAM TRANCE VNN1	ASAM BChE DAN IGFBP-5 IL-20 NOV NTB-A PRL Siglec-3 TIM-3 TNF R1 uPAR		
AIN-93G/soy	CXCL16 RSTN	VLDL R RSTN	E-selectin GAL-3 LCN2	bFGF GAL-3 GAL-4 GDF-9 EPHB1 IL-22 Ra1 JAM-A RGM-C SEMA-3C SR-A1

^a^ Raw data is available in the Appendix A. ^b^ Protein names and functions are listed in Appendix A. Red color indicates increased expression.

**Table 6 ijms-26-06137-t006:** Protein expression in mouse plasma as a function of diet.

Array ^a^	Protein ^b^	*Fmr1^HET^* Female	*Fmr1^KO^* Female	WT Male	*Fmr1^KO^* Male
Q4	IL-17F		↑ 5.7-fold casein		
Q4	IL-20		↑ 37-fold casein		
Q5	bFGF				↑ 1.1-fold soy
	TNF R1		↑ 1.3-fold casein		
Q6	JAM-A				↑ 1.1-fold soy
Q8	PSPN	↑ 6.7-fold casein			
	SLAM	↑ 7.4-fold casein			
	TRANCE	↑ 44-fold casein			
	EPI	↑ 6.2-fold casein			
Q9	BAFF		↑ *-fold casein		
Q12	ASAM		↑ 2.6-fold casein		
Q13	SEMA3C				↑ 5.5-fold soy
	Siglec-3		↑ 6.1-fold casein		
	SR-AI				↑ 1500-fold soy
	TIM-3		↑ 260-fold casein		
Q14	VLDL R		↑ 8.8-fold soy		
Q15	BCHE		↑ 1.7-fold cas		
Q16	CA2	↑ 2.7-fold casein			
	NTB-A		↑ 5.5-fold casein		
	VNN-1				↑ 3.1-fold soy
Q17	GAL-4				↑ 18-fold soy
	EPHB1				↑ 6.9-fold soy
	GDF-9				↑ 3.5-fold soy
	IL-22 Ra1				↑ 19-fold soy
	NTN-4	↑ 7.4-fold casein			
Q19	ANG	↑ 4.2-fold casein			
	BID	↑ 1.4-fold casein			

^a^ Raw data is available in the Appendix A. ^b^ Protein names and functions are listed in Appendix A. * Fold change cannot be calculated as one group has a mean of 0 pg/mL. Casein refers to AIN-93G/casein and soy refers to AIN-93G/soy diet. ↑ refers to an increase in fold change.

**Table 7 ijms-26-06137-t007:** Comparison of fold change in protein expression in mouse plasma as a function of casein versus soy-based pelleted infant formula (IF) and AIN-93G diets.

Protein ^a^	WT IF ^b^	*Fmr1^KO^* IF ^b^	WT AIN ^c^	*Fmr1^KO^* AIN ^c^	Studies Concur
B7-1	0.73 (*p* ≤ 0.05)		0.49 (*p* = 0.15)		--
CD36	2.37 (*p* ≤ 0.05)		1.15 (*p* = 0.46)		--
ICAM-1	3.68 (*p* ≤ 0.05)		0.74 (*p* = 0.47)		--
IGFBP5		0.73 (*p* ≤ 0.05)		0.63 (*p* = 0.04)	YES
LEP	0.11 (*p* ≤ 0.02)		1.11 (*p* = 0.93)		--
MFG-E8	4.34 (*p* = 0)		1.14 (*p* = 0.81)		--
MMP-10	2.04 (*p* ≤ 0.04)		1.10 (*p* = 0.94)		--
NEP	0.25 (*p* ≤ 0.05)		0.94 (*p* = 0.84)		--

^a^ Protein names and functions are in Appendix A. ^b^ CIF versus SIF [12]. ^c^ AIN-93G/casein versus AIN-93G/soy.

**Table 8 ijms-26-06137-t008:** Top protein targets after Benjamini–Hochberg’s correction.

Tissue	Protein ^a^	*Fmr1^HET^* Female	*Fmr1^KO^* Female	WT Male	*Fmr1^KO^* Male
Cortex	CD14	↑casein vs. soy			
Hippocampus	CD4				↑ soy vs. casein
	IL-15 Ra		↑ casein vs. soy		
			↑ vs. male KO (casein)		
	KLB		↑ casein vs. soy		
			↑ vs. male KO (casein)		
	PRSS34	↑ casein vs. soy			
		↑ vs. male KO (casein)			
Hypothalamus	NOV	↑ vs. male WT (soy)			
Plasma	ASAM		↑ vs. KO male (casein)		
	EPI	↑ casein vs. soy			
		↑ vs. KO male (casein)			
	GAL-3				↑ vs. HET (soy)
	GDF-9				↑ vs. male WT (soy)
					↑ vs. fem KO (soy)
	JAM-A				↑ vs. fem KO (soy)
	LCN2			↑ vs. HET (soy)	
	MBL-2	↑ vs. WT (casein)			
	PRL		↑ vs. male KO (casein)		
	RSTN	↑ vs. WT (soy)			
		↑ vs. male KO (soy)			
	SEMA3C				↑ soy vs. casein
					↑ vs. HET (soy)
					↑ vs. fem KO (soy)
	SLAM	↑ vs. fem KO (casein)			
	SR-A1				↑ soy vs. casein
	VNN1	↑ vs. male KO (casein)			

^a^ Protein names and functions are found in Appendix A. Casein refers to AIN-93G/casein and soy refers to AIN-93G/soy diet. ↑ refers to an increase in fold change.

## Data Availability

The original contributions presented in this study are included in the article/Appendix A. Further inquiries can be directed to the corresponding author.

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
