# Peer review of "Soy Protein Isolate Affects Blood and Brain Biomarker Expression in a Mouse Model of Fragile X"

_ijms, 2025, doi:10.3390/ijms26136137_

Round 1
Reviewer 1 Report
Comments and Suggestions for Authors
The authors examined differential protein expression the brain, and blood of the fragile X mouse line using a commercial subcontract for the study. There are several major concerns with this study in regards to the study design, animal numbers, and how the results are presented and discussed. These are outlined below.
Major concerns/suggestions:
- In the results (section 2.1), the authors list a number of proteins that are differentially-expressed (referring to Table 1), but no significance statements are made, nor are the values for the differential expression provided. There is no way to assess the significance and magnitude of the differential expression. Table 1 only refers to supplementary data, but not the specific one in 19 files. When one goes to the supplementary data, there are no figure legends and it is impossible to tell what the green versus pink data points refer to—which are the control versus the supplemented diet?
- Every table and section afterwards continues these concerns. Additionally, there continues to be a lack of clarity on the direction, and magnitude of the differential expression. For example, the sentence “Specifically, Prss34 and TRANCE were differentially expressed in Fmr1HET females” line 11 does not specify the two comparisons being made. Are these proteins higher in Fmr1HET females in general, or when they are on the diet, and higher compared to which other group?
- Why are HET females used as controls instead of WT females? Is this because of the breeding scheme used? This explanation also seems odd as where did the WT males come from then? The concerns for a HET females for an X-linked gene is that there would be chimerism in the HET females, and as such they might not serve as real controls. While I understand that the HET and KO females are littermates, I am not sure that justifies the lack of WT females for this study.
- The authors are aware, and discuss that the female mice were sacrificed without regards to estrous cycle. This is a major concern as many genes differ with cycle. In addition, time of day for the euthanizing can also result in differences between animals.
- How were these brain sections consistently isolated for each animal? Was a brain matrix used? Measurements (bregma) from points, or specific markers on the brain to ensure consistency with brain areas between animals?
- For the HET females and WT males on casein diet (assuming this, as the graphs in the supplement are not labeled with a figure legend), there were only N =4 and N=5 and in most cases, there was a cluster of expression with some outliers. These outliers may actually be the reason that there are significant differences, especially with low “N” in the study.
- What is the rationale for examining/comparing KO females on one diet to a WT male on a different diet…
- Are there diet effects regardless of sex? Or sex effects regardless of diet?
- Figure 3 is missing a key for the colors versus diet.
- Is Figure 4 only male mice? The figure legend where it says “ie WT and fmr1KO male mice…” does not make it clear if this is just an example of the cohorts or if this figure only shows results for males. This figure does give the figure key for diet, which is needed for the supplemental data as well.
- For all of the arrays, it is not clear how they were standardized. What protein or proteins were used as controls across all diets and tissues and sexes?
Minor comments:
- Abstract: line 20, AIN-93G/cas diet has not been previously defined before first use in the abstract.
- The discussion about soy use in standard chow diets across studies (which is now in the discussion) actually could be moved to the introduction because this argument is an important one for mouse studies everywhere.
- Throughout the rest of the manuscript abbreviations are not adequately defined in the data. I.e. use of the abbreviation “cas” is never defined (although one would assume that is the casein diet). This might seem minor but it means that the figures and tables do not stand alone. It is also never made clear that casein is the control diet versus the soy diet. Why not call this ‘control’.
- Table 1: protein nomenclure for rodent genes uses the first letter capital and the remaining letters lower case i.e. Ccl6.
- Why are the array numbers highlighted? Aren’t these just arbitrary numbers by the company? Are any proteins represented more than once on different arrays, and if so, do any of the differentially-expressed signals appear on more than one array?
- The table of the protein names and function can be a supplemental table.
Reviewer 2 Report
Comments and Suggestions for Authors
In this manuscript, Boeck and coworkers using the well-established model for Fragile X syndrome, the Fmr1 knockout mice, investigate how soy protein isolate versus casein diets affect biomarker expression in blood and brain tissues, cortex, hippocampus, and hypothalamus. Aiming to uncover implications for the pathophysiology and biomarkers of FXS they examine gene-diet interactions, sex differences, and tissue-specific effects. Based on previous evidence suggesting that diet alters the propensity for epileptic activity, the neurodevelpment and the gut-brain singaling processes in this animal model of FXS, authors try to understand how soy protein isolate affects biomarkers in Fmr1 KO mice. They conclude that diet with soy protein isolate only affects the expression of several brain and blood proteins as a function of Fmr1 genotype and sex.The manuscript is generally well written. The methods used are appropriate, and the presentation of results is clear and detailed, and the conclusions are supported by the data. The manuscript adds new information to the current literature. However, there is room for improvement, and below I propose some suggestions.
1). Table 8 is very long and dense, containing dozens of proteins that disrupt the readability of the tex. I think it should be more suitably placed in the supplementary material. Authors could also use color-coded tables to show the direction and significance of protein changes across tissues, genotypes, diets, and sexes.
2). The current graphs provide more information than the simple mean ± SEM typically shown in column graphs. However, it would be preferable to replace the current graph with a box plot, which conveys more statistical detail and is generally easier to interpret. This is particularly advisable for small sample sizes as in the present study.
3). Figure 3. A legend for color interpretation is missing.
4). Authors should summarize their main findings at the beginning of the Discussion to enhance readability and consolidation of relatively complex results.
Reviewer 3 Report
Comments and Suggestions for Authors
In the manuscript titled "Soy Protein Isolate Affects Blood and Brain Biomarker Expression in a Mouse Model of Fragile X," a team led by Boeck and Westark investigates the levels of protein biomarkers in relation to FMRP levels, sex, and diets based on matched casein and soy protein isolate in FMR1 knockout (KO) and littermate mice.
Overall, the authors present several interesting points. However, there are a couple of weaknesses that need to be addressed.
1.The authors found several differentially expressed proteins in various tissues, and they may want to verify the expression of some of these proteins through Western blot analysis.
2. Additionally, it would be beneficial to explore any potential protein-protein interactions among the differentially expressed proteins.
3. What are the clinical implications of the changes in protein expression? Are there mechanisms involved that could guide future research? The authors should consider discussing these aspects in more detail.
Round 2
Reviewer 1 Report
Comments and Suggestions for Authors
This is a revision of a paper that I reviewed previously. The authors have done a good job in revising the figures and figure legends in the paper to make the findings easier to read and understand. However, three major concerns still exist and were not adequately responded to by the authors. First, there were a low number of animals, without a POWER analysis to determine the magnitude of the difference that could be detect with the numbers available. It is not clear if there might be missing findings due to this, or findings that are interpreted by the outliers (from estrous cycle or circadian differences between individual animals) and not real differences. Second, the authors state that they dissected tissues, but did not provide methodology for repeatability of the study. Third, none of the identified proteins were validated during a secondary analysis. Secondary validation is a standard procedure for gene, and protein array expression analysis.
Specific comments are as follows:
- In response to my comment that genes and proteins can change with estrous cycle and time of day, the authors simply state that estrous cycle stage was not measured. This really makes the data—especially with so few female mice uninterpretable. The authors state also that they did take the mice all at the same time of day, but do not provide what time the euthanasia was done in the methods. Again, time of day can influence circadian levels of proteins, but also can be relative to the rodent feeding cycle—were they eating or had been eating prior to sacrifice, or were they in their sleep phase?
- In response to my question about how the brain sections were isolated, they authors stated that they did not use a brain matrix or bregma measurements. They say the “entire hypothalamus and left cortex” were dissected. The authors need to provide details as to how the “entire hypothalamus” was identified—visually? Under a dissecting microscope? How was the hypothalamus dissected away from the thalamus consistently?
- I do not agree that N=3 is a minimum for animal studies. Was a POWER analysis done to confirm this? Our Power analysis usually finds N=8-10 per group is necessary for gene expression (QPCR) fold differences.
- In response to reviewer #2’s question about protein-protein interaction, the authors respond that it is an interesting question but would have to be done with a new cohort. The authors could provide in silico string analysis to look at the interactome for the identified proteins. This is a standard analysis after gene or protein arrays.
- Reviewer #2 points out that secondary confirmation is needed for the identified proteins. I agree (and should have also pointed that out in my original review). Without secondary confirmation of the findings in an independent group of animals, the findings are not validated, and potentially represent outliers rather than true responses to the diets.
Reviewer 3 Report
Comments and Suggestions for Authors
The author addressed all the comments and the manuscript was much improved. No additional comments.
Author Response
The author addressed all the comments and the manuscript was much improved. No additional comments.
Reply: Thank you.
Round 3
Reviewer 1 Report
Comments and Suggestions for Authors
The authors have adequately responded to the reviews, and have improved the paper significantly. The increased methodological detail provides for repeatability, and the critique of their shortfalls improves the soundness of the conclusions. The additional STRING analysis provides some interesting biological pathways to consider for future experiments.